# Telomere Dysfunction in Pediatric Patients with Differences/Disorders of Sexual Development

**DOI:** 10.3390/biomedicines12030565

**Published:** 2024-03-02

**Authors:** Haifaou Younoussa, Macoura Gadji, Mamadou Soumboundou, Bruno Colicchio, Ahmed Said, Ndeye Aby Ndoye, Steffen Junker, Andreas Plesch, Leonhard Heidingsfelder, Ndeye Rama Diagne, Alain Dieterlen, Philippe Voisin, Patrice Carde, Eric Jeandidier, Radhia M’kacher

**Affiliations:** 1Cell Environment DNA Damage R&D, Genopole, 91000 Evry-Courcouronnes, France; younoussa.haifaou@ucad.edu.sn (H.Y.); ahmed.said11.as@gmail.com (A.S.); philippe.voisin@cell-environment.com (P.V.); 2Service of Biological Hematology & Oncology-Hematology (BHOH), National Centre of Blood Transfusion (NCBT), Faculty of Medicine, Pharmacy and Odonto-Stomatology (FMPO), Cheikh Anta Diop University of Dakar (UCAD), Dakar BP 5002, Senegal; 3Medical Diagnostic Laboratory, Diamniadjo Hospital, UFR-Santé de Thiès, Iba Der Thiam University of Thies, Thiès BP A967, Senegal; mamadou.soumboundou@univ-thies.sn (M.S.); prndeyerama.diagne@univ-thies.sn (N.R.D.); 4IRIMAS, Institut de Recherche en Informatique, Mathématiques, Automatique et Signal, Université de Haute-Alsace, 68093 Mulhouse, France; bruno.colicchio@uha.fr (B.C.); alain.dieterlen@uha.fr (A.D.); 5Service de Chirurgie Pédiatrique de L’hôpital Albert Royer, Cheikh Anta Diop Ave, Dakar BP 25755, Senegal; ndeyeaby.ndoye@ucad.edu.sn; 6Institute of Biomedicine, University of Aarhus, DK-8000 Aarhus, Denmark; sjunker@biomed.au.dk; 7MetaSystems GmbH, D-68804 Altlussheim, Germany; aplesch@metasystems.de (A.P.); lheidingsfelder@metasystems.de (L.H.); 8Department of Hematology, Gustave Roussy Cancer Campus, 94805 Villejuif, France; dr.pcarde@gmail.com; 9Service de Génétique, Groupe Hospitalier de la Région de Mulhouse Sud-Alsace, 68100 Mulhouse, France; jeandidiere@ghrmsa.fr

**Keywords:** DSDs, chromosomal aberrations, telomere shortening, telomere dysfunctions

## Abstract

Differences/Disorders of sex development (DSDs) are conditions in which the development of chromosomal, gonadal, and anatomical sexes is atypical. DSDs are relatively rare, but their incidence is becoming alarmingly common in sub-Saharan Africa (SSA). Their etiologies and mechanisms are poorly understood. Therefore, we have investigated cytogenetic profiles, including telomere dysfunction, in a retrospective cohort of Senegalese DSD patients. Materials and methods: Peripheral blood lymphocytes were sampled from 35 DSD patients (mean age: 3.3 years; range 0–18 years) admitted to two hospital centers in Dakar. Peripheral blood lymphocytes from 150 healthy donors were used as a control. Conventional cytogenetics, telomere, and centromere staining followed by multiplex FISH, as well as FISH with *SRY*-specific probes, were employed. Results: Cytogenetic analysis identified 19 male and 13 female patients with apparently normal karyotypes, two patients with Turner syndrome, and one patient with Klinefelter syndrome. Additional structural chromosome aberrations were detected in 22% of the patients (8/35). Telomere analysis revealed a reduction in mean telomere lengths of DSD patients compared to those of healthy donors of similar age. This reduction in telomere length was associated with an increased rate of telomere aberrations (telomere loss and the formation of telomere doublets) and the presence of additional chromosomal aberrations. Conclusions: To the best of our knowledge, this study is the first to demonstrate a correlation between telomere dysfunction and DSDs. Further studies may reveal the link between telomere dysfunction and possible mechanisms involved in the disease itself, such as DNA repair deficiency or specific gene mutations. The present study demonstrates the relevance of implementing telomere analysis in prenatal tests as well as in diagnosed genetic DSD disorders.

## 1. Introduction

Differences/Disorders of sex development (DSDs) are congenital anomalies characterized by atypical chromosomal, gonadal, and anatomical sex development resulting in ambiguous external and internal genitalia and hormonal dysfunction [1,2]. The term “disorders of sexual development” is currently used to replace terms such as “sexual ambiguity”, “intersex”, “hermaphroditism”, or “pseudo hermaphroditism”. This new terminology was related essentially not only to the potential pejorative of the old [3] but also associated with the new classification during the Pediatric Endocrine Society and European Society for Pediatric Endocrinology (LWEPS-ESPE) conference [1,4]. This classification is based on the chromosomal analysis and clinical features [5,6]. Nevertheless, DSD classification remains very difficult because similar phenotypes can have multiple etiologies [4,7]. Currently, the management of patients is multidisciplinary, involving imaging, genetics, and hormonology [8].

DSDs are rare genetic disorders with incidences varying between 1/4500 and 1/5500 live births worldwide [9,10,11,12]. DSD 46,XX is the most represented variant [13] (1/14,000 to 1/15,000 vs. 1/20,000 for DSD 46,XY) [14,15,16]. However, DSD incidence is becoming alarmingly common in Africa in general (1/3000 in Egypt) and in sub-Saharan Africa (SSA) (1/357 in Ghana), in particular [12,17]. This increased incidence could be related not only to a highly endogenous and inbred population, to the efficacy of prenatal diagnosis in these countries but also to environmental factors. The lack of epidemiological studies and specific structures, as well as cultural barriers, make the treatment and follow-up of these diseases in Africa very difficult.

Unfortunately, major challenges with the diagnosis and management of DSD patients persist in this part of the world. In addition, the etiologies and specific biomarkers related to DSDs remain poorly understood. 

DSD patients exhibit a very high risk of gonadal cancers [18], hypogonadism [19,20], lung and breast cancers [21,22], as well as various fertility complications and hormonal insufficiency [23]. Chromosomal instability, a driving force of the progression of malignancy, has been previously described in DSD patients [24]. Previous studies have shown that cells derived from patients with trisomy 13 (Patau syndrome), trisomy 18 (Edwards syndrome), trisomy 21 (Down syndrome), or monosomy X (Turner syndrome) exhibit a significantly higher frequency of sporadically acquired non-specific whole chromosome losses and gains compared to control cases [25,26]. It has also been reported that patients with DSDs have chromosomal aberrations that are often related to the Y chromosome [27]. 

Telomeres are nucleoprotein complexes located at the ends of eukaryotic chromosomes, and they have a critical role in preserving chromosomal integrity and stability [28,29]. Telomere length is used as a biomarker of biological age [30] and an aging-disease risk factor [31,32,33]. Telomere dysfunction is related to chromosomal instability, either through progressive telomere shortening or telomere aggregation and telomere loss and deletion [34,35]. The loss of telomere functionalities is considered the one major mechanism for the progression of genomic instability [36]. Significant shortening of telomere length and significantly higher frequencies of telomere loss and deletion have been found in peripheral lymphocytes of patients with cancer and genetic diseases compared to healthy donors of the same age [37]. Chromosomal instability is also associated with telomere shortening and loss of telomere functionality that ultimately leads to end-to-end chromosomal fusions, thus contributing to the initiation and progression of cancer [38,39,40]. 

In this study, cytogenetic analysis has been conducted to evaluate not only structural and numerical chromosomal aberrations but also telomere profiles of DSD patients from SSA. We demonstrate for the first time that telomere instability is a common characteristic of SSA DSD patients. Telomere instability could indeed be playing a role in the formation of additional chromosomal aberrations. To our knowledge, this is the first study to address telomere profiles in a cohort of DSD patients. We hypothesize that chromosomal aberrations in these patients are related to telomere dysfunction. 

## 2. Materials and Methods

### 2.1. Declaration of Ethics

This study and research protocol were approved by the ethics committee of the Cheikh Anta Diop University of Dakar (Protocol 041512019/CER/UCAD). Patients and their guardians or family members were included in the study only after receiving detailed information about the study and signing an informed consent form. Data were collected and processed in a confidential manner.

### 2.2. Pediatric Patients

We conducted a retrospective study of 35 pediatric patients with DSD admitted to clinical consultation from November to December 2021 at the Diamniadio Children’s Hospital and the Albert Royer Pediatric Surgery Department of FANN. These are two large hospitals in Dakar that cover the entire Dakar region and surrounding areas. The patients were initially received for clinical examination in the two above-mentioned hospitals and referred to the National Centre of Blood Transfusion (NCBT) in Dakar. Sampling and cytogenetic analyses were performed at the NCBT. All patients had disorders of sexual development ranging from mild hypospadias pubertal delay to overt external genitalia ambiguity.

### 2.3. Culture of Lymphocytes, Preparation of Metaphases, and Analysis of Karyotypes after G-Banding (GTG-Banding)

Peripheral blood lymphocytes were cultured for 72 h, and standard GTG-Banding was performed according to a previous publication [41]. A total of 50 metaphases from each sample were analyzed at the resolution level of 550 bands. Karyotypes are presented according to ISCN2020 [42].

### 2.4. Detection of SRY by Fluorescence In Situ Hybridization (FISH)

Fluorescence in situ hybridization (FISH) was performed on interphase cells harvested from freshly collected whole blood and on cultured PHA-stimulated cells. The slide was washed in 2x standard citrate saline (2x SSC) at 37 °C for 30 min. The slides were dehydrated in three alcohol gradients, 70%, 90%, and 100%, for 2 min each and then air dried. In total, 10 μL of the probe was deposited on the slides and then covered with a coverslip. The slides with coverslips were placed on a thermobrite (ThermoFisher, Illkirch, France) for denaturation at 76 °C for 7 min and then hybridized at 37 °C for 24 h. After 24 h, the slides were washed in 1xPBS solution to loosen the coverslips, then immersed in a solution of 0.4x SSC with 0.3% NP-40 for 2 min at 73 °C and in a solution of 2x SSC with 0.1% NP-40 at room temperature for 2 min. Cells were counterstained with 4′,6-diamidino-2-phenylindole (DAPI) solution and then rinsed in PBS before mounting the slides with vectashield orp-phenylenediamine (PPD). Two specific probes for the SRY gene were used: the Vysis AneuVysion probe (Vysis LSI SRY/CEP X FISH Probe Kit, Abbott, Des Plaines, IL 60018, USA) and the Cytocell probe (SRY Probe, Cytocell Aquarius, Symex, Bremerhaven, Germany). The Vysis probe consists of two parts: The Xp11.1-q11.1 CEP X (DXZ1) Spectrum Green part specific for the X chromosome and the Yp11.3 LSI SRY Spectrum Orange part specific for the SRY gene. The Cytocell probe is a mixture of three probes: One probe (SRY), Yp11.31, in red (Texas Red), a control probe for the Y chromosome (DYZ1), Yq12 (heterochromatic block) in green (Green) and a control probe for the X centromere (DXZ1), Xp11.1-q11.1 in blue (Aqua). Images were captured using an automated acquisition module Autocapt software (MetaSystems, version 3.9.1) using an automated ZEISS Plan-Apochromat 63×/1.40 oil and CoolCube 4 Digital High-Resolution CCD camera. The analysis was performed on 200 cells for each sample [25].

### 2.5. Staining of Telomere and Centromere Sequences

Telomere and centromere staining were performed using a Cy-3-labelled PNA probe specific for telomere sequences and a FITC-labelled PNA probe specific for centromere sequences (Cell Environment, Evry, France), as previously described [37]. 

### 2.6. Telomeres Length Analysis

Telomere quantification was performed on interphase cells using TeloScore software (Cell Environment, version 1.1.2, Evry, France). Quantitative image acquisition was performed using MetaCyte software (MetaSystem, version 3.9.1, Altlussheim, Germany) and a ZEISS Plan-Apochromat (Zeiss, Oberkochen, Germany) and CoolCube 1 Digital High-Resolution CCD Camera (MetaSystems, Altlussheim, Germany). The exposure and gain settings remained constant between captures. The mean fluorescence intensity (FI) of telomeres was automatically quantified in 10,000 nuclei on each slide. The quantifications were performed on triplicate slides. Telomere length, measured as the mean FI, correlates strongly with telomere length measured by conventional Southern blot analysis using the telomeric restriction fragment (TRF) (R^2^ = 0.721 and *p* = 2.128 × 10^−8^). The mean telomere length is expressed in kb.

### 2.7. Scoring of Telomere Aberrations

Analysis of metaphase spreads allowed the detection of telomere abnormalities using ChromoScore Software (Cell Environment, version 1.1.2, Evry, France). The images of metaphases were captured using the automated acquisition module Autocapt software (MetaSystems, version 3.9.1) and a ZEISS Plan-Apochromat 63×/1.40 oil (Zeiss, Oberkochen, Germany) and CoolCube 1 Digital High-Resolution CCD Camera (MetaSystems, Altlussheim, Germany) with constant settings for exposure and gain. 

Telomere abnormalities scored were (i) sister telomere loss, likely occurring in G2, and defined as a telomere signal-free end at a single chromatid [27], (ii) telomere deletion defined as the loss of two telomere signals on the same chromosome arm (likely resulting from the loss of one telomere in G1/S), an aberration considered to represent double-strand breaks, leading to the activation of DNA damage response. Automatic scoring of these aberrations was performed using ChromoScore software (Cell Environment, version1.1.2, Evry, France). An operator validated and excluded the falsely recorded aberrations.

### 2.8. Multicolor FISH (M-FISH Technique)

The M-FISH technique employs multicolor probes that make it possible to identify each of the 22 pairs of autosomes as well as the X and Y chromosomes by “painting” them with individual colors. Moreover, fragments of chromosomes translocated into non-homologous chromosomes were also identified using M-FISH. 

After telomere quantification and the automatic capture of metaphases with telomere and centromere staining, the slides were washed in 2x SSC for 30 min at 70 °C. After rinsing with 0.1x SSC, the slides were denatured using NaOH and subsequently washed with 0.1x SCC and 2x SSC and sequentially dehydrated in 70%, 95%, and 100% ethanol and air-dried. After denaturation of the M-FISH probe (M-FISH 24XCyte, Metasystems, Altlussheim, Germany) for 5 min at 75 °C, the probe was added to the slides and incubated at 37 °C for two days. The slides were subsequently rinsed with 0.4x SSC for 2 min at 72 °C and then with 2x SSC/0.005% (Tween-20). The slides were counterstained with DAPI and mounted in PPD. 

### 2.9. Statistical Analysis

Data were analyzed using the Wilcoxon-Mann–Whitney rank sum test (comparison of two sub-groups) or the Kruskal–Wallis non-parametric test (comparison of three sub-groups). We tested the null hypothesis that the sub-groups are considered identical populations. A *p*-value < 0.05 is considered statistically significant to reject the null hypothesis.

## 3. Results

### 3.1. Clinical Profile of DSD Patients

This study was performed on 35 retrospective DSD pediatric patients admitted to two hospital centers. Inclusion criteria for patients in this study were age less than 18 years and congenital malformation according to clinical examination. The mean age of these patients at diagnosis was 2.62 years (0–14 years) and 3.33 years (0–18 years) at cytogenetic analysis. Twelve of these DSD patients (34.5%) had non-classified pathologies. Clinical features of all patients are listed in Table 1.

### 3.2. Conventional and Molecular Cytogenetic Investigations

After PHA stimulation of freshly isolated circulating lymphocytes and of cells in culture, conventional and molecular cytogenetics were performed to retrieve chromosomal abnormalities. Table 2 summarizes the results of G-Banding and *SRY*-specific FISH.

Using conventional cytogenetics, 20 patients had a male karyotype profile (46,XY) without apparent chromosomal abnormalities (Figure 1A). Thirteen patients had a female karyotype (46,XX) without apparent chromosomal abnormalities (Figure 1B). Two patients, YH008 and YH015 (Table 2) (Appendix A), had mosaic Turner syndrome 46,XX[12]/45,X[5] and 45,X[28]/46,XY[20], respectively (Figure 1C). 

Molecular analysis using co-hybridization with FISH probes specific for *SRY* and X centromere sequences, respectively, was used to validate the conventional cytogenetic data (Figure 2). A male profile with corresponding signals was found in 19 patients (Figure 2A). The female profile with corresponding signals was found in 13 patients (Figure 2B). We have also confirmed the Turner syndrome profile for two patients (YH008 and YH015) detected by conventional cytogenetics (Figure 2C). In addition, Klinefelter syndrome was also found in patient YH009 (Figure 2D). The latter anomaly had not been identified using conventional cytogenetics. 

Using telomere and centromere-specific probes, several additional structural chromosome aberrations were identified in metaphases from eight of the patients (22%), such as a dicentric chromosome (Figure 3A) and acentric chromosomes with chromosome deletions (2 patients) (Figure 3B). Telomere fusions (3 patients) (Figure 3C) were observed in addition to chromosome fragmentations (2 patients) (Figure 3D), as well as chromosomal breaks (3 patients).

### 3.3. Quantification of Telomere Length of DSD Patients

To understand the origin and mechanisms underlying the formation of these additional aberrations, we assessed the telomere lengths of circulating lymphocytes of the DSD patients using an automated approach based on cytogenetic preparations and FISH (Aging kit Cell Environment). This approach permits not only the assessment of mean telomere length but also the intercellular variation and proportion of cells with extreme telomere shortening (<5 kb) in vast numbers of interphase nuclei (Figure 4A). A large cohort of 150 healthy donors (0.5–79 years of age) served as controls in this study. Telomere length in healthy donors was age-dependent and characterized by high inter-individual variation (R^2^ = 0.316 and *p* = 2.48 × 10^−10^) (Figure 4B). The spontaneous decrease in telomere length in healthy donors was 79 bp per year. Interestingly, there was a significant difference (*p* < 10^−6^) between the mean telomere length of DSD patients and that of healthy donors of similar age, being 6.99 kb (4.33–9.85 kb) for DSD patients and 10.5 kb (5.2–13.9 kb) for healthy donors, respectively (Figure 4C). 

### 3.4. Telomere Dysfunction of DSDs Patients

Telomere dysfunction relates to any telomere structural aberration that effectively abolishes the presence of a functional telomere, resulting in chromosome end-to-end fusion, dicentric chromosome formation, and ongoing chromosomal instability. In metaphases of DSDs patients, telomere loss (Figure 5A) was significantly more extensive than in those of healthy donors of similar age 1.42 (range 0–8) per cell vs. 0.52 (range 0–3.23) per cell, (*p* < 10^−7^), respectively (Figure 5B). In contrast, telomere doublet formation was significantly lower in DSD patients (0.8, range 0–1.7 per cell) than in healthy donors (5.51, range 1.29–12.17 per cell) (*p* < 10^−10^), respectively (Figure 5C). In addition, high inter-individual variation in the frequencies of telomere losses and doublets was recorded in DSD patients (Figure 5D). 

A closer inspection of the data on telomere aberrations in individual chromosomes revealed that many chromosomes, especially chromosomes 21 and 22, were more frequently affected than others (Figure 6).

Furthermore, we quantitated telomere length and the frequency of telomere losses and telomere doublet formations in DSDs patients with or without additional chromosome aberrations to assess a putative correlation between telomere dysfunction and the formation of chromosomal aberrations. Telomere loss in patient cells with additional chromosome aberrations (1.60 telomere loss/cell) was not significantly higher than in those with normal karyotypes (1.32 telomere loss/cell). Of note, dicentric chromosomes, a driving force of chromosomal instability, were identified in two patients with structural chromosome aberrations. Interstitial telomeres were observed in dicentric chromosomes, demonstrating the role of telomere dysfunction in their formations. 

In addition, we observed a high frequency of micronuclei in these DSDs, which is relevant given that micronuclei can originate from those with additional chromosomal aberrations and chromosomal pulverizations. The chromosomal and telomere profiles of each patient are listed in Table 3, in addition to clinical features.

## 4. Discussion

Rapid and precise diagnosis of DSDs patients is considered a major public health and societal challenge, particularly in light of the increasing prevalence of DSDs in, e.g., SSA countries in general and in Senegal in particular. Our study is a first step in the design of novel biomarkers for the identification and diagnosis of DSDs patients in order to optimize patient care. Furthermore, it will contribute to our understanding of mechanisms involved in the development of DSDs. Currently, the analysis of karyotypes is the primary and most accurate approach to diagnoseDSDs in clinical settings. In the present study, we have extended the classical cytogenetic analyses by including, for the first time, the analysis of telomeres to assess the telomere and chromosomal profiles in a Senegalese cohort of 35 DSDs patients. Our study has highlighted the implication of telomere dysfunction in the development of DSDs, thus adding a novel biomarker for the identification and diagnosis of DSD patients.

The high prevalence of DSDs in SSA countries is ascribed to two major causes: (i) consanguinity and endogamy in these populations [2], and (ii) the excessive use of insecticides such as Dichlorodiphenyltrichloroethane (DDT) in agriculture or farming in order to eradicate tropical endemic diseases [43]. Consanguinity and endogamy are well-known causes of many genetic disorders, including DSDs [44]. DDT is recognized as an endocrine disruptor causing a reduction in sperm quality and increased risk of congenital diseases [45]. Accumulating evidence suggests that prenatal exposure to DDT is considered a risk factor in the incidence of DSDs. The presence of chromosomal aberrations in DSDs patients could be associated with genotoxic stress.

Conventional cytogenetic analysis of our cohort identified two patients with mosaic Turner syndrome (45,X/46,XY, and 45,X/46,XX). Turner syndrome is a chromosomal DSD caused by the monosomy of the X chromosome [46]. The mosaic subtypes of Turner syndrome are relatively rare [47]. We confirmed our findings by employing *SRY*-specific probes. By this approach, we also identified a mosaic Klinefelter syndrome (KS) (46,XY/47,XXY), which had not been detected by conventional cytogenetic, possibly because only 20 metaphases are being analyzed for conventional karyotyping, but 200 nuclei with the automated FISH approach. KS is the most frequently observed chromosomal DSD, with an estimated frequency of 1/500 to 1/1000 [48]. The mosaic subtype accounts for some 10–20% of KS cases [49].

Using conventional and molecular cytogenetics, structural chromosomal aberrations were identified in only three of the DSD patients (3/35; 8.5%). This rate is less than that previously described in other registers and in other countries (varying between 13% and 15%), thus underscoring a significant contribution of the etiology to the occurrence of DSD. Our findings highlight the limitations of using conventional karyotyping in clinical genetics and call for new genetic tests to identify additional biomarkers characteristic for these diseases [50].

For a reliable and precise analysis of additional chromosomal aberrations in DSD patients, we employed telomere and centromere staining. Using that approach, we identified additional non-clonal chromosomal aberrations in 22% of the DSD patients, including dicentric chromosomes, DNA pulverization, acentric chromosomes, terminal deletions, and chromatid breaks. Dicentric chromosomes are considered the best biomarker for irradiation-induced DNA damage as well as for chromosomal instability [37]. In our cohort, we detected a dic(Y;4) with interstitial telomere sequences, indicating that the formation of this dicentric chromosome is related to telomere dysfunction. Several DSDs studies have previously reported the implication of the Y chromosome in the formation of a dicentric [51,52,53,54,55,56]. In addition, chromosome pulverization was observed in two patients in our cohort as the consequence of telomere dysfunction and probably of micronuclei formation and chromothripsis mechanisms [57]. We demonstrate that our protocol for telomere and centromere staining offers improved detection of all chromosomal aberrations, clonal and non-clonal ones. Employment of this technique in SSA cytogenetic laboratories will constitute a major step forward in the management of DSDs patients.

Analysis of telomere lengths and telomere aberrations revealed shortening of and accumulation of aberrations of telomeres in the DSDs patients. A link between reduced fertility and telomere dysfunction has been reported previously [38,58]. Reduced telomere length has been described in patients with chromosomal DSDs, including KS associated with acute lymphoblastic leukemia (ALL). It was suggested that telomere dysfunction in these KS patients may contribute to the pathogenesis of ALL [24]. However, no studies have addressed a direct link between decreased telomere length and the prevalence of DSDs.

Here, we demonstrate for the first time a significant age-independent reduction in mean telomere length in our cohort of DSDs patients compared to that observed in healthy donors of similar age. Our data reveal an accelerated aging process in DSD patients, thus opening new horizons in our understanding of these disorders in terms of possible mechanisms.

In addition to telomere length, we studied telomere aberrations involving losses and doublet formations. The frequency of telomere aberrations in the DSDs cohort was also independent of age. However, we recorded a more frequent loss of telomeres than the occurrence of doublets in contrast to the healthy controls. The analysis of telomere aberrations in each chromosome revealed a higher rate of these aberrations in chromosomes 21 and 22 compared to other chromosomes. These findings call for further investigation.

In this paper, we have conducted a comprehensive and in-depth cytogenetic study, including telomere analysis of patients with DSD from West Africa, in particular from Senegal. Although this study suffers from limitations in terms of the complexity of DSDs, the difficulty in precise diagnosis, and the role of a specific environment, it nevertheless allowed us to gain more insight into African DSDs that may present distinct genetic features related to the environment and the management methods in these regions. However, analysis of a large prospective cohort with access to complete clinical and biological parameters is required to validate the present results and, thus, the application of these techniques for future genetic diagnosis of DSDs. The techniques employed for cytogenetic diagnosis in the present cohort are accessible, and their application is feasible in Senegal. Hence, the development of these techniques will be an indispensable tool for the management of DSDs, which still constitute a major challenge in hospitals today.

## 5. Conclusions

The results of this study have allowed the establishment of a complete cytogenetic diagnosis, including telomere analysis, and also defined specific features of DSDs that have not been previously reported. Indeed, our data demonstrate for the first time that telomere shortening and telomere aberrations represent one of the most common cytogenetic features of DSDs. The results show a major involvement of small acrocentric chromosomes, especially chromosomes 21 and 22, in telomere aberrations, allowing us to better understand the mechanisms resulting in DSDs. The sequential analysis of telomere length and aberrations for DSDs patients may contribute to our knowledge and better understanding of molecular mechanisms of the accelerated aging process and the implication of DNA repair mechanisms and specific mutations in these diseases.

## Figures and Tables

**Figure 1 biomedicines-12-00565-f001:**
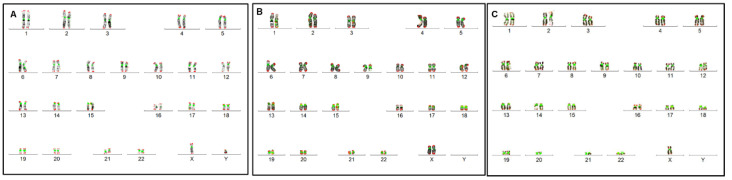
G-banded karyotypes supplemented with centromere and telomere-specific FISH. (**A**) Normal male karyotype (46,XY) from a DSD patient with ambiguous genitalia but no apparent chromosomal abnormalities. (**B**) Normal female karyotype (46,XX) from DSDs patients with ambiguous genitalia without visible chromosomal abnormalities. (**C**) Karyotype of a DSD patient with ambiguous genitalia and a monosomy X (45,X, Turner syndrome).

**Figure 2 biomedicines-12-00565-f002:**
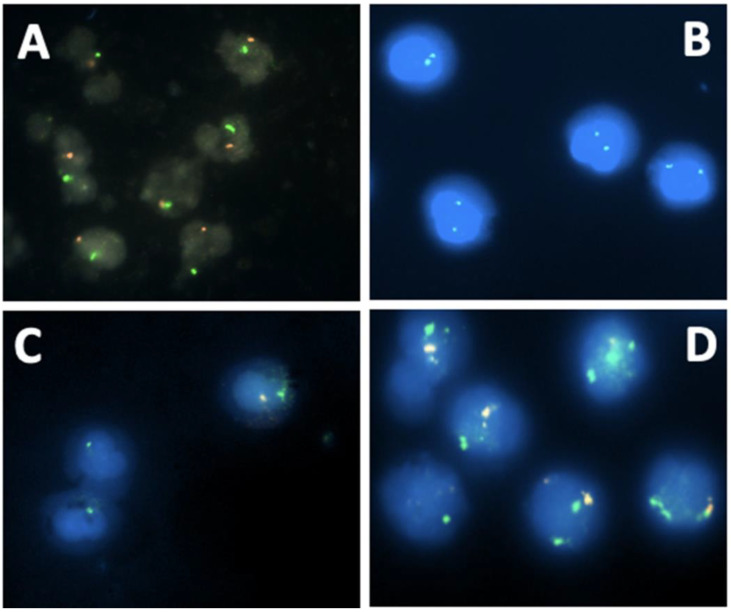
DSDs profiles using molecular cytogenetics with specific probes for *SRY* (red signal) and X centromere (green signal) (**A**) Normal FISH signals of a male DSD patient (one green signal and one red signal); (**B**) Normal FISH signals of female DSD patient (two green signals); (**C**) Mosaic male Turner syndrome. Two cell populations: one with a red signal (appears whitish) and a green signal, and the other with only a green signal due to the loss of the Y chromosome; (**D**) Mosaic Klinefelter syndrome. Two cell populations: one with one green signal and one red signal (normal cells) and the second cell population with two green signals and one red signal.

**Figure 3 biomedicines-12-00565-f003:**
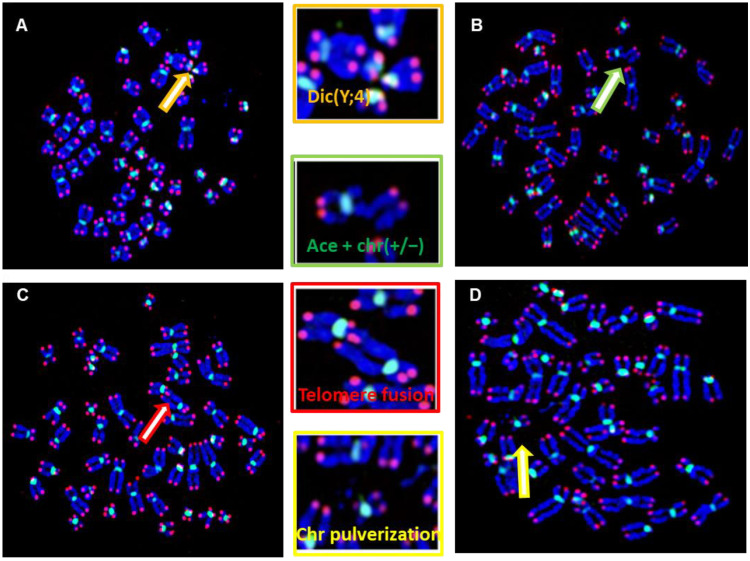
Structural chromosome aberrations in DSD patients were identified by telomere (red signal) and centromere (green signal) staining. (**A**) Metaphase showing a dicentric chromosome, dic(Y;4) between chr 7 with interstitial telomeres, demonstrating that telomere fusion is the origin of the formation of aberration. (**B**) Double Strand breaks (DSB) resulting in the formation of an acentric chromosome and chromosome deletion (ace(+/−) with chr(+/−)). These aberrations have been observed in 4 patients (YH014, YH020, YH032, YH033). (**C**) Telomere fusion between two different chromosomes (**D**) DNA pulverization observed in two patients (YH016 and YH017). Metaphase of a DSD with DNA fragmentation observed in patients YH004 and YH033. (**C**) Chromatid break observed in several DSDs involving different chromosomes. (**D**) Metaphase showing a fusion of two chromosomes from the chromatids of each chromosome with interstitial telomeres, observed in DSDs (YH016, YH017).

**Figure 4 biomedicines-12-00565-f004:**
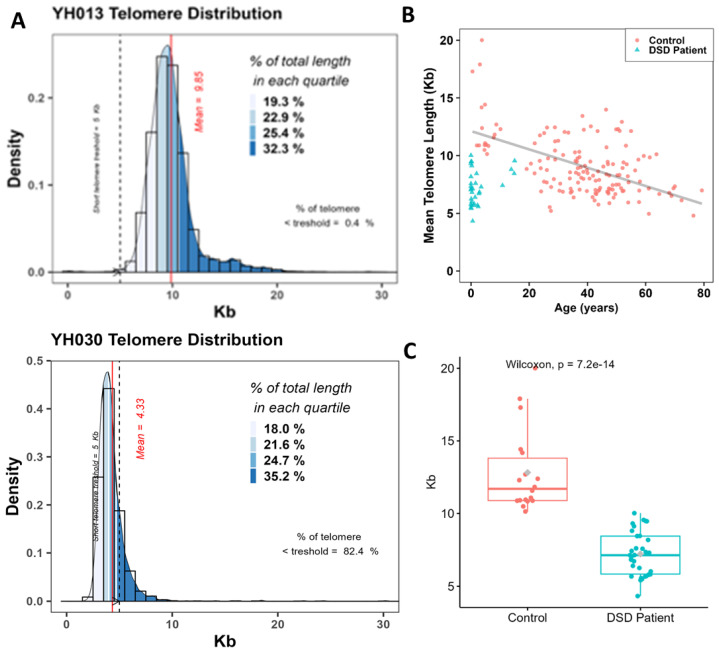
Quantification of telomere length in circulating lymphocytes of DSDs patients. (**A**) Distribution of the telomere lengths in circulating lymphocytes of patients YH013 and YH030 that exhibit the most extreme telomere lengths in our DSDs cohort. The mean telomere length is presented by the red line and the frequency of cells with extreme telomere shortening (<5 kb) is presented as a dashed blue line. The different quartiles of fluorescence signal intensities in telomeres are also shown. (**B**) Telomere length (kb) as a function of age in lymphocytes from healthy donors (150 donors, mean age 36 years, range 0.5–79 years) (red circles) and in DSDs patients (blue triangle). (**C**) A significant difference between the means of telomere length of DSD patients and those from healthy donors with similar ages.

**Figure 5 biomedicines-12-00565-f005:**
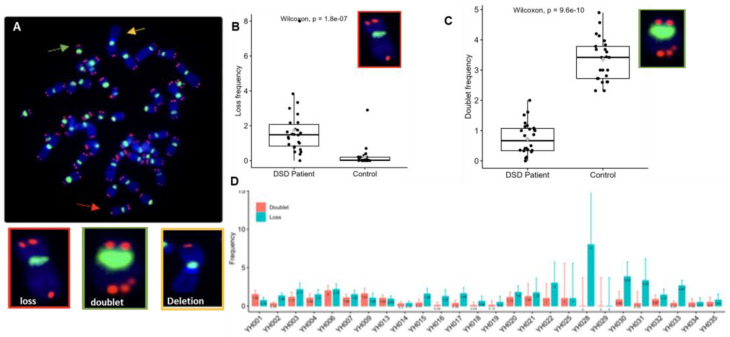
Telomere aberrations are identified by assessing telomeres on individual chromosomes in metaphases of circulating lymphocytes. (**A**) Representative metaphase after telomere (red) and centromere (green) staining showing different types of telomere aberrations: telomere loss (red arrow), telomere doublet (green arrow), and telomere deletion (yellow arrow). (**B**,**C**) Frequencies of telomere losses and telomere doublet formations per cell in DSDs patients and in the control panel with similar age. A significant increase in the frequency of telomere loss was observed in DSDs patients in comparison to the control group (*p* < 1.8 × 10^−7^). A significant decrease in telomere doublet formation in DSDs patients was observed compared to that in the control (*p* < 9.6 × 10^−10^). (**D**) The frequency of telomere loss and telomere doublets in each patient shows high inter-individual variation. One hundred metaphases were analyzed per patient.

**Figure 6 biomedicines-12-00565-f006:**
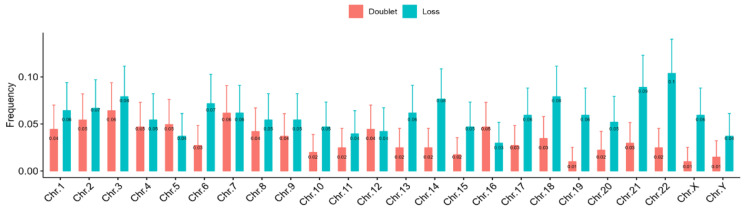
The pooled frequencies of telomere aberrations (loss and doublet) per chromosome of all the DSDs patients reveal that the most abundant aberrations are in chromosomes 3, 14, 21, and 22.

**Table 1 biomedicines-12-00565-t001:** Clinical characteristics of Differences/Disorders of sex development (DSDs) patients before cytogenetic investigations.

Characteristics	No. of Patients
DSDs	35
Assigned sex	
Male	17
Female	13
ND	5
Age (years)	
At diagnosis	2.6
At analysis	3.5
Type	
Clitoral hypertrophy	8
Micropenis + cryptorchidism	5
Isolated hypospadias	4
Isolated cryptorchidism	2
Isolated micropenis	1
Hypogonadism + gynecomastia	1
Short stature + Turner syndrome	1
Other DSDs with diverse congenital malformations	12

ND: The sex of the patient could not be determined by clinical examination.

**Table 2 biomedicines-12-00565-t002:** Cytogenetic profiles of DSDs patients.

Cytogenetic Profile	No. of Patients
Nb of analyzed DSDs patients	35
Conventional cytogenetics	35
Molecular cytogenetic (SRY)	35
Karyotype results	
46,XY	18
46,XX	14
46,XX[12]/45,X[5]	1
45,X[28]/46,XY[20]	1
nuc ish(DXZ1x2,SRYx1)[85/200]	1
Structural chromosome aberrations	8

**Table 3 biomedicines-12-00565-t003:** Clinical and cytogenetic characteristics of the used DSDs cohort.

ID	Assigned SEXE	Age Month	Reason for Consultation	FISH (*SRY*)	Karyotypes	Telomere Length (kb)	Telomere Loss/Cell	Telomere Doublet/Cell	Additional Chromosome Aberrations
YH001	M	0	Micropenis and cryptorchidism	XY (*SRY*+)	46,XY	9.31	0.78	1.52	
YH002	M	178	Micropenis and cryptorchidism)	XY (*SRY*+)	46,XY	8.5	1.35	0.35	
YH003	F	3	Prader Type I	XX (*SRY*−)	46,XX	9.15	2.17	1.17	
YH004	M	0	ND	XY (*SRY*+)	46,XY	7.66	1.5	1.05	DNA fragmentation Dicentric with interstitial telomeres
YH005	M	4	Clitoral hypertrophy	XX (*SRY*−)	46,XX	6.32	NA	NA	
YH006	M	33	ND	XY (*SRY*+)	46,XY	7.57	2.19	2	
YH007	M	19	Hypospadias	XY (*SRY*+)	46,XY	8.09	1.52	1.08	
YH008	M	120	Micropenis and cryptorchidism	28% XY/57% X (*SRY*+)	45,X[28]/46,XY[20]	6.09	NA	NA	
YH009	M	7	Hypospadias	ish(DXZ1x2,SRYx1)[85/200] (*SRY*+)	46,XY[115]/47,XXY[85]	8.21	1.06	1.71	
YH010	F	42	Clitoral hypertrophy	XX (*SRY*−)	46,XX	7.29	0	0	
YH011	M	1	ND	XY (*SRY*+)	46,XY	5.42	0	0	
YH012	M	166	Hypogonadism and gynecomastia	XY (*SRY*+)	46,XY	8.12	0	0	
YH013	F	3	Clitoral hypertrophy	XY (*SRY*+)	46,XY	9.85	0.94	1.03	
YH014	F	0	Clitoral hypertrophy	XX (*SRY*−)	46,XX	9.47	0.37	0.3	Chromosomal breakage 46,XX,del(15)(q10q26)
YH015	F	180	Statural retardation and Turner syndrome	71% XX/29% X (*SRY*−)	46,XX[12]/45,X[5]	9.55	1.59	0.42	
YH016	ND		ND	XX (*SRY*−)	46,XX	5.56	1.28	0.09	chromatid fusion of ch 8 and 10
YH017	M	13	ND	XY (*SRY*+)	46,XY	5.83	1.65	0.35	chromatid fusion of ch 15 and 4 chromatid fusion of ch 15 and 9
YH018	ND	0	ND	XY (*SRY*+)	46,XY	6.71	0.6363636364	0.09090909091	
YH019	ND	0	ND	XX (*SRY*−)	46,XX	8.18	0.5	0.125	
YH020	M	130	Hypospadias	XY (*SRY*+)	46,XY	7.42	1.8	1.13	Chromosome 2 breakage
YH021	ND	0	ND	XY (*SRY*+)	46,XY	5.5	1.75	1.25	
YH022	M	27	Micropenis and cryptorchidism	XY (*SRY*+)	46,XY	5.7	3	1	
YH023	F	23	Clitoral hypertrophy	XX (*SRY*−)	46,XX	6.34	NA	NA	
YH024	M	11	Micropenis and cryptorchidism	XY (*SRY*+)	46,XY	5.67	NA	NA	
YH025	F	17	external Genitalia Anomaly	XX (*SRY*−)	46,XX	6.39	1	1	
YH026	F	0	Clitoral hypertrophy	XX (*SRY*−)	46,XX	7.02	NA	NA	
YH027	F	0		XX (*SRY*−)	46,XX	5.8	NA	NA	
YH028	ND	0	Polymalformation	XY (*SRY*+)	46,XY	6.42	8	0	
YH029	M	1	ND	XY (*SRY*+)	46,XY	6.25	NA	NA	
YH030	M	7	Ovotestis	XX (*SRY*−)	46,XX	4.33	3.83	0.8	Acentric chromosome
YH031	F	30	Prader Type II	XY (*SRY*+)	46,XY	5.57	3.333	0.33	
YH032	M	1	hypospadias	XY (*SRY*+)	46,XY	7.14	1.47	0.87	Chromosome breakage
YH033	F	35	ND	XX (*SRY*−)	46,XX	6.84	2.67	0.374	DNA Fragmentation
YH034	F	3	ND	XX (*SRY*−)	46,XX	6.01	0.5	0.42	
YH035	F	2	Micro penis	XY (*SRY*+)	46,XY	5.68	0.8	0.5	

## Data Availability

The data presented in this study are available on request from the corresponding author.

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
