# Peer review of "Telomere Dysfunction in Pediatric Patients with Differences/Disorders of Sexual Development"

_biomedicines, 2024, doi:10.3390/biomedicines12030565_

Round 1
Reviewer 1 Report
Comments and Suggestions for Authors
I would like to thank the authors for giving me the opportunity to review their interesting manuscript. This report details a cohort of children with DSDs and the results of examining their cytogenetic features. Overall, I found the manuscript interesting though I had a number of comments/suggestions:
- My first comment is rather simple but, the authors (I don't think) make it clear from the beginning why they decided to conduct telomere studies. It appears that the original aim was to characterize the cytogenetics of patients and then after noting telomere abnormalities, the decision was made to go back and do telomere studies on the entire cohort. It would perhaps be good to provide a better framework of the aims of the study in the Abstract and Introduction.
Introduction:
- I would suggest using the official definition as the current language in the manuscript ("disruption of chromosomal, gonadal and anatomical sex harmony completeness and balance") does not seem appropriate.
- Line 56, what do the authors mean when they say some DSDs are "better managed"?
- There is no framework given for the different type of DSDs (ie: XX DSD, XY DSD, sex chromosome abnormalities, etc.). Without this, it is somewhat difficult to follow the rest of the introduction
- Paragraph 2: This discussion on the risks for tumors is hard to understand (eg: what are "type II germline cancers") and I think attempts to condense a lot of information into a short paragraph. I gather the authors are trying to make a connection between the risk for cancers and chromosome/telomere instability though but I think these paragraphs (2, 3, 4) can be better structured to get this point across.
Materials and Methods:
- For the section titled "pediatric patients", what was the exact inclusion criteria - ie: how was a disorder of sexual differential defined? The authors mention including patients with "pubertal delay" which is not usually included in this category.
- In addition to this, the authors later break down the "sex" of patients but how was this defined? Was this the sex assigned at birth or the sex the parents/patients chose to use?
- Page 4, Lines 164-166: Should this paragraph be in the next section (title "Staining of telomere and centromere sequences")
Results:
- Worryingly, 12 patients are listed who had "non-defined pathologies" - what does this mean exactly? Without knowing these patients' features (either phenotypic or cytogenetic), it is hard to justify why they were even included in the study.
- I would strongly recommend moving the the table summarizing the clinical and cytogenetic characteristics into the main manuscript and expanding the column titled "reason for consultation" to add more details - having simple DSD is not adequate.
- I don't see the utility in including Figures 1A, 1B and 1C or Figure 1 for that matter. In addition, why do the authors think karyotype did not identify mosaic Kleinfelter syndrome in patient YH009?
- The authors appear to have identified a number of chromosomal aberrations however none of these are given in the standard ISCN nomenclature. These should all be given in the correct format (eg: loss of a long arm of chromosome 15 would be something like 46,xx,del(15)(q10q26))
- Were the study populations' telomere lengths compared to the entire 150 person cohort or only the age-matched controls? It appears that the authors did both from Figures 4B and 4C. Was this to show how short they were - ie: comparable to much older individuals? Also I would suggest the authors include the actual mean telomere length for each subject in the manuscript somewhere - either in a table or combined into one of the other tables
Discussion:
-The authors' assertion that the current study could significantly improved the genetic diagnosis of DSD patients is somewhat misleading. There are numerous causes for telomere dysfunction and there is no data given on the overall rate of telomere dysfunction in infants from this region. In addition, if this rate of consanguinity and endogamy is elevated in this population, it is possible that there are many other genetic abnormalities present in these infants which may be contributing to their clinical disease. Standard karyotyping (with appropriate SRY FISH analyses) is of course the first step in the diagnosis of any DSD patient, but if such analyses are unrevealing, next generation sequencing panel or exome/genome sequencing are now commonly utilized and are becoming much cheaper and more available broadly. These allow for a more precise molecular diagnosis which is generally more helpful when it comes to ongoing management. Generally, if the authors are linking telomere dysfunction to the risks for cancer and, therefore, changes to management based on that knowledge, this needs to be stated more clearly. Otherwise, it appears that there is more research needed to understand if telomere dysfunction actually causes the physical development of a DSD or is merely a by-product which increase the risks for malignancy.
- Line 368 - Is the incidence of DSD particularly high in Senegal? The authors have not presented data to suggest this is so.
Comments on the Quality of English LanguageNo issues with the English language
Author Response
I would like to thank the authors for giving me the opportunity to review their interesting manuscript.This report details a cohort of children with DSDs and the results of examining their cytogenetic features. Overall, I found the manuscript interesting though I had a number of comments/suggestions:
- My first comment is rather simple but, the authors (I don't think) make it clear from the beginning why they decided to conduct telomere studies. It appears that the original aim was to characterize the cytogenetics of patients and then after noting telomere abnormalities, the decision was made to go back and do telomere studies on the entire cohort. It would perhaps be good to provide a better framework of the aims of the study in the Abstract and Introduction.
Thank you to the reviewer for this comment
Indeed, karyotyping and cytogenetic investigation is part of clinical setting in DSD patients. The aim of this work is to find other biomarkers for a better stratification of patients. We have developed cytogenetic techniques and protocols based on telomere and centromere staining which makes it possible to analyze chromosomal aberrations as well as telomere status.
The abstract and the introduction have been modified, and the framework of the study is defined better.
Introduction:
- I would suggest using the official definition as the current language in the manuscript ("disruption of chromosomal, gonadal and anatomical sex harmony completeness and balance") does not seem appropriate.
The definition has been modified and reference has been added.
- Line 56, what do the authors mean when they say some DSDs are "better managed"?
This sentence has been modified
- There is no framework given for the different type of DSDs (ie: XX DSD, XY DSD, sex chromosome abnormalities, etc.). Without this, it is somewhat difficult to follow the rest of the introduction
A paragraph has been added in the different types of DSD
- Paragraph 2: This discussion on the risks for tumors is hard to understand (eg: what are "type II germline cancers") and I think attempts to condense a lot of information into a short paragraph. I gather the authors are trying to make a connection between the risk for cancers and chromosome/telomere instability though but I think these paragraphs (2, 3, 4) can be better structured to get this point across.
Thank you to the reviewer, this paragraph has been structured
Materials and Methods:
- For the section titled "pediatric patients", what was the exact inclusion criteria - ie: how was a disorder of sexual differential defined? The authors mention including patients with "pubertal delay" which is not usually included in this category.
The inclusion criteria of patients have been defined in this paragraph
- In addition to this, the authors later break down the "sex" of patients but how was this defined? Was this the sex assigned at birth or the sex the parents/patients chose to use?
It is the sex assigned at birth
- Page 4, Lines 164-166: Should this paragraph be in the next section (title "Staining of telomere and centromere sequences")
The paragraph has been deleted
Results:
- Worryingly, 12 patients are listed who had "non-defined pathologies" - what does this mean exactly? Without knowing these patients' features (either phenotypic or cytogenetic), it is hard to justify why they were even included in the study.
Thank you for this comment, the sentence has been modified, these are DSD patients with unclassified pathologies, demonstrating the complexity of this disease. The sentence has been modified
- I would strongly recommend moving the table summarizing the clinical and cytogenetic characteristics into the main manuscript and expanding the column titled "reason for consultation" to add more details - having simple DSD is not adequate.
Table has been moved and the reasons of consultation have been added in some cases.
- I don't see the utility in including Figures 1A, 1B and 1C or Figure 1 for that matter. In addition, why do the authors think karyotype did not identify mosaic Kleinfelter syndrome in patient YH009?
The reason has been added, karyotype has been performed on 20 and 50 metaphases. However, the screening of the presence of SRY and X centromere was performed on 200 nuclei.
- The authors appear to have identified a number of chromosomal aberrations however none of these are given in the standard ISCN nomenclature. These should all be given in the correct format (eg: loss of a long arm of chromosome 15 would be something like 46,XX,del(15)(q10q26))
The nomenclature has been respected
- Were the study populations' telomere lengths compared to the entire 150 person cohort or only the age-matched controls? It appears that the authors did both from Figures 4B and 4C. Was this to show how short they were - ie: comparable to much older individuals? Also I would suggest the authors include the actual mean telomere length for each subject in the manuscript somewhere - either in a table or combined into one of the other tables.
The mean telomere length and the rate of cells with drastic telomere shortening have been added in the supplementary data. In this version of the manuscript, we have introduced all the data in the mean text
Discussion:
-The authors' assertion that the current study could significantly improved the genetic diagnosis of DSD patients is somewhat misleading. There are numerous causes for telomere dysfunction and there is no data given on the overall rate of telomere dysfunction in infants from this region. In addition, if this rate of consanguinity and endogamy is elevated in this population, it is possible that there are many other genetic abnormalities present in these infants which may be contributing to their clinical disease. Standard karyotyping (with appropriate SRY FISH analyses) is of course the first step in the diagnosis of any DSD patient, but if such analyses are unrevealing, next generation sequencing panel or exome/genome sequencing are now commonly utilized and are becoming much cheaper and more available broadly. These allow for a more precise molecular diagnosis which is generally more helpful when it comes to ongoing management. Generally, if the authors are linking telomere dysfunction to the risks for cancer and, therefore, changes to management based on that knowledge, this needs to be stated more clearly. Otherwise, it appears that there is more research needed to understand if telomere dysfunction actually causes the physical development of a DSD or is merely a by-product which increase the risks for malignancy.
Thank you very much for this comment, we agree with the reviewer concerning the diagnostic strategy of DSD: Standard karyotype and genomic analysis.
In industrialized countries, where all diagnostic procedures, including genetic testing, are covered by the health insurance system, this approach is easy to apply.
In this study, we have begun to apply a reliable and easily transferable technique to Sub Africa countries for the detection of chromosomal aberrations. We will work on the genomic approach to define a panel of genes specific to this population that could be introduced into the clinic at very low cost. This work is on progress and will be the subject of another article.
- Line 368 - Is the incidence of DSD particularly high in Senegal? The authors have not presented data to suggest this is so.
Indeed, the incidence is very high, but the lack of epidemiological studies and associated publications has prevented us from the notification of this rate of incidence in the Senegal in this article.
Reviewer 2 Report
Comments and Suggestions for Authors
Thank you for requesting our review of your manuscript titled “Telomere dysfunction leads to chromosomal aberrations in pediatric patients with disorders of sexual development” in the Special Issue: Telomere Biology in Human Health, Aging and Diseases.
The premise of demonstrating correlation between telomere dysfunction and the various chromosomal disorders using telomere analysis is a very good one and does merit study. The concept of designing a genetic test for identification and treatment is noteworthy but quite premature at this juncture. This paper is very intriguing and quite deserving of further consideration. There are a number of issues that must be addressed, however, before consideration for publication. Most considerable is the size of the cohort, as 35 subjects is a small sample size but this is further confounded by 12 subjects without diagnosis. Therefore this paper has insufficient data to draw any meaningful conclusions. The authors would do best to consider the 27 subjects as a pilot study, at best. Therefore, we recommend that the manuscript be rejected at this time.
Title:
· Line 2: “leads to” does not reflect scientific correlation.
Abstract:
· Line 25: Replacement of the term “Disorders of sex development (DSD)” with more acceptable terminology, such as “sex chromosome disorders or aneuploidies” or “children with X and Y chromosomal variations”, etc., should be considered throughout the manuscript.
Introduction:
· Line 50: The range of these incidence ratios is inaccurate and contradicts ratios mentioned in Lines 381 and 386. They need to be corrected.
· Line 61: The gonadal tumor information is outdated and not necessary or appropriate detail for the scope of this manuscript.
Materials and Methods:
· This section is strong and detailed. It is well written and designed.
Results:
· Line 235-236: “12 of these DSD patients (34.5%) had non-defined pathologies” What are ‘non-defined pathologies”? They comprise a large percentage of this small cohort and further clarification is needed.
· Line 241: A side-by-side comparison of resultant profiles using conventional vs molecular cytogenetics would be interesting, if space permits. There is need for table or figure that exposes the increase of detailed and novel results in comparison to the conventional method.
In conclusion, this paper does have great potential for future research and study. The idea is both novel and innovative, however, it has major flaws that need addressing before publication may be considered. The data gathered needs more consideration and how it will be analyzed. The 12 subject with “non-defined pathologies” is major flaw and needs more definition and clarification before this paper can be considered for publication.
Author Response
Thank you for requesting our review of your manuscript titled “Telomere dysfunction leads to chromosomal aberrations in pediatric patients with disorders of sexual development” in the Special Issue: Telomere Biology in Human Health, Aging and Diseases.
The premise of demonstrating correlation between telomere dysfunction and the various chromosomal disorders using telomere analysis is a very good one and does merit study. The concept of designing a genetic test for identification and treatment is noteworthy but quite premature at this juncture. This paper is very intriguing and quite deserving of further consideration. There are a number of issues that must be addressed, however, before consideration for publication. Most considerable is the size of the cohort, as 35 subjects is a small sample size but this is further confounded by 12 subjects without diagnosis. Therefore this paper has insufficient data to draw any meaningful conclusions. The authors would do best to consider the 27 subjects as a pilot study, at best. Therefore, we recommend that the manuscript be rejected at this time.
Title:
- Line 2: “leads to” does not reflect scientific correlation.
The title has been modified
Abstract:
- Line 25: Replacement of the term “Disorders of sex development (DSD)” with more acceptable terminology, such as “sex chromosome disorders or aneuploidies” or “children with X and Y chromosomal variations”, etc., should be considered throughout the manuscript.
We have used the terminology adapted in the Chicago consensus in 2006. The definition of DSD has been modified
Introduction:
- Line 50: The range of these incidence ratios is inaccurate and contradicts ratios mentioned in Lines 381 and 386. They need to be corrected.
The incidence has been corrected
- Line 61: The gonadal tumor information is outdated and not necessary or appropriate detail for the scope of this manuscript.
The paragraph has been rewritten
Materials and Methods:
- This section is strong and detailed. It is well written and designed.
The section has been modified
Results:
- Line 235-236: “12 of these DSD patients (34.5%) had non-defined pathologies” What are ‘non-defined pathologies”? They comprise a large percentage of this small cohort and further clarification is needed.
The DSD pathology has been defined but the classification is not possible. The sentence has been modified
- Line 241: A side-by-side comparison of resultant profiles using conventional vs molecular cytogenetics would be interesting, if space permits. There is need for table or figure that exposes the increase of detailed and novel results in comparison to the conventional method.
The table with all the data has been added
In conclusion, this paper does have great potential for future research and study. The idea is both novel and innovative, however, it has major flaws that need addressing before publication may be considered. The data gathered needs more consideration and how it will be analyzed. The 12 subject with “non-defined pathologies” is major flaw and needs more definition and clarification before this paper can be considered for publication.
Thank you for the reviewers for the comments. Major modifications have been made according the reviewers suggestions.
Round 2
Reviewer 1 Report
Comments and Suggestions for Authors
I thank the authors for giving me the opportunity to review their manuscript once again and for their careful addressing of my previous concerns. I have the following comments/suggestions:
- I thank the authors for including more information on the inclusion critieria. It sounds like patients were referred to this study and NOT examined personally by the study team. Hence, there was no specific information provided on their phenotype other than "concern for DSD". If that is accurate, I would ask the authors to include an explanation like this. RIght now, it is not appropriate to simply say "patients had non-classifed pathologies" without more context.
- In Table 1, do the authors use "structural retardation" to mean short stature? If so, I would use that term instead of the former to be more accurate.
- I would also add in Table 1 some notation that "Male", "Female" or "ND" are assigned sex
Comments on the Quality of English LanguageN/A
Author Response
I thank the authors for including more information on the inclusion critieria. It sounds like patients were referred to this study and NOT examined personally by the study team. Hence, there was no specific information provided on their phenotype other than "concern for DSD". If that is accurate, I would ask the authors to include an explanation like this. RIght now, it is not appropriate to simply say "patients had non-classifed pathologies" without more context.
- In Table 1, do the authors use "structural retardation" to mean short stature? If so, I would use that term instead of the former to be more accurate.
The modification has been added
- I would also add in Table 1 some notation that "Male", "Female" or "ND" are assigned sex
Table 1 has been modified
Reviewer 2 Report
Comments and Suggestions for Authors
Abstract:
Line 26: the phrase “of chromosomal, gonadal and anatomical sexis atypical” should be changed to “of chromosomal, gonadal, and anatomical sexes is atypical.”
Line 40: The phrase "To the best of our knowledge this study is the first" should be changed to "To the best of our knowledge, this study is the first"
Introduction:
Line 76: This paragraph about cancers is irrelevant to the purpose of your manuscript and should be omitted unless a clearer connection is made between the paper's purpose and cancer risk.
Author Response
Line 26: the phrase “of chromosomal, gonadal and anatomical sexis atypical” should be changed to “of chromosomal, gonadal, and anatomical sexes is atypical.”
The correction has been included
Line 40: The phrase "To the best of our knowledge this study is the first" should be changed to "To the best of our knowledge, this study is the first"
The change has been made
Introduction:
Line 76: This paragraph about cancers is irrelevant to the purpose of your manuscript and should be omitted unless a clearer connection is made between the paper's purpose and cancer risk.
The paragraph has been modified